# Neural tracking of speech mental imagery during rhythmic inner counting

**Lingxi Lu[1,2], Qian Wang[3], Jingwei Sheng[2], Zhaowei Liu[2], Lang Qin[2,4], Liang Li[5], Jia-Hong Gao[1,2,6]***

[1]PKU-IDG/McGovern Institute for Brain Research, Peking University, Beijing, China; [2]Center for MRI Research, Academy for Advanced Interdisciplinary Studies, Peking University, Beijing, China; [3]Department of Clinical Neuropsychology, Sanbo Brain Hospital, Capital Medical University, Beijing, China; [4]Department of Linguistics, The University of Hong Kong, Hong Kong, China; [5]Speech and Hearing Research Center, School of Psychological and Cognitive Sciences, Peking University, Beijing, China; [6]Beijing City Key Lab for Medical Physics and Engineering, Institution of Heavy Ion Physics, School of Physics, Peking University, Beijing, China

**Abstract** The subjective inner experience of mental imagery is among the most ubiquitous human experiences in daily life. Elucidating the neural implementation underpinning the dynamic construction of mental imagery is critical to understanding high-order cognitive function in the human brain. Here, we applied a frequency-tagging method to isolate the top-down process of speech mental imagery from bottom-up sensory-driven activities and concurrently tracked the neural processing time scales corresponding to the two processes in human subjects. Notably, by estimating the source of the magnetoencephalography (MEG) signals, we identified isolated brain networks activated at the imagery-rate frequency. In contrast, more extensive brain regions in the auditory temporal cortex were activated at the stimulus-rate frequency. Furthermore, intracranial stereotactic electroencephalogram (sEEG) evidence confirmed the participation of the inferior frontal gyrus in generating speech mental imagery. Our results indicate that a disassociated neural network underlies the dynamic construction of speech mental imagery independent of auditory perception.
DOI: https://doi.org/10.7554/eLife.48971.001

*For correspondence:
jgao@pku.edu.cn

**Competing interests:** The authors declare that no competing interests exist.

## Introduction

People can generate internal representations in the absence of external stimulation, for example, by speaking or singing in their minds. This experience is referred to as mental imagery, which reflects a higher cognitive function of the human brain (*Kosslyn et al., 2001*). Speech mental imagery is among the most pervasive human experiences, and it has been estimated that people spend more than one-quarter of their lifetime engaged in such inner experiences (*Heavey and Hurlburt, 2008*). Many functional magnetic resonance imaging (fMRI) studies have reported distributed neural networks involving the left pars opercularis in the inferior frontal cortex (Broca's area) (*Aleman et al., 2005*; *Kleber et al., 2007*; *Papoutsi et al., 2009*; *Koelsch et al., 2009*; *Price, 2012*; *Tian et al., 2016*), the premotor cortex (*Price, 2012*; *Kleber et al., 2007*; *Koelsch et al., 2009*; *Tian et al., 2016*; *Aleman et al., 2005*), the inferior parietal lobe (*Kleber et al., 2007*; *Tian et al., 2016*), the middle frontal cortex (*Tian et al., 2016*), the anterior part of the insula (*Koelsch et al., 2009*; *Tian et al., 2016*; *Aleman et al., 2005*; *Ackermann and Riecker, 2004*), and the superior temporal gyrus (*Aleman et al., 2005*) that can be activated by speech mental imagery. However, limited by the temporal resolution of the fMRI technique, the neural dynamics of speech mental imagery have been mostly overlooked.

Neuronal information processing is shaped by multiscale oscillatory activity (*Buzsáki and Draguhn, 2004*), in which subjective experience in consciousness involves ongoing neural dynamics on the time scale of hundreds of milliseconds (*Dehaene and Changeux, 2011*). Therefore, establishing a neurophysiological model of speech mental imagery is highly important for elucidating the spectral-temporal features of imagery-induced neural activity. However, in the experimental environment, tracking subjects' internal subjective representations in milliseconds remains highly challenging due to the lack of external signals identifying imagery-related neural activity. Considering the limitations of the current methodologies, several neurophysiological studies have used the imagery-perception repetition paradigm to measure the sensory attenuation effect caused by speech mental imagery on subsequent auditory perception (*Tian and Poeppel, 2013*; *Tian and Poeppel, 2015*; *Ylinen et al., 2015*; *Whitford et al., 2017*; *Tian et al., 2018*). Preliminary magnetoencephalography (MEG) studies have shown that the modulatory effect of imagined speech on the amplitude of event-related responses to a subsequent auditory stimulus with matched pitch and/or content occurs as early as 100 to 200 ms (*Tian and Poeppel, 2013*; *Tian and Poeppel, 2015*; *Ylinen et al., 2015*), and in an electroencephalogram (EEG) study, N1 suppression was observed when a matched phoneme was generated covertly before an audible phoneme was presented (*Whitford et al., 2017*). Recently, *Tian et al. (2018)* examined the influence of imagined speech on the perceived loudness of a subsequent syllable and demonstrated the involvement of the sensory-driven cortex (i.e. the auditory cortex) in the early-stage interaction of speech imagery and perception. These neurophysiological findings implicate the early involvement of sensory-related neural substrates in the process of speech mental imagery, while in the adaptation paradigm, the neural pathway of top-down mental imagery cannot be disentangled from the bottom-up perception. To date, the neural dynamics of mental imagery independent of auditory perception remain puzzling.

Frequency-tagging is a promising approach for disentangling exogenous and endogenous processes when the corresponding neural activity changes periodically at different frequencies. Recently, frequency-tagging has been used to characterize higher-lever processes such as music rhythmic perception (*Nozaradan et al., 2011*; *Tal et al., 2017*; *Lenc et al., 2018*; *Nozaradan et al., 2018*) and speech linguistic construction (*Ding et al., 2016*; *Sheng et al., 2019*). For example, *Nozaradan et al. (2011)* asked eight musicians to imagine a binary/ternary meter when listening to a constant beat and found spectral responses not only to the beat stimulation but also to the imagined meters of this beat. *Ding et al. (2016)* reported cortical responses to internally constructed linguistic structures such as words, phrases and sentences at different frequencies during the comprehension of connected speech. Therefore, frequency-tagging is an excellent candidate as an approach for capturing the dynamic features during the internal construction of speech mental imagery by tagging the corresponding neural activity in the frequency domain.

Here, we apply the frequency-tagging method in a behavioral task involving speech mental imagery to experimentally isolate the neural manifestations of internally constructed speech from the stimulus-driven processes. In this way, we are able to directly track the neural processing time scales of speech mental imagery in the frequency domain (in our experiment, we manipulated the imagery-rate frequency to 0.8 Hz). Furthermore, we used the MEG source estimation method to localize the neural underpinnings of the isolated processes of speech imagery independent of the bottom-up auditory perception and confirmed the localization of the neural sources by intracranial stereotactic electroencephalogram (sEEG) recordings. We hypothesized that in addition to the sensory-driven auditory cortex, distinctive neural networks are responsible for generating speech mental imagery.

## Results

### Cortical tracking of speech mental imagery

In the MEG study, we aimed to identify the cortical representations of speech mental imagery independent of stimulus-driven responses. We constructed a 4 Hz sequence of auditory stimuli that were mentally organized into 0.8 Hz groups by a speech mental imagery task (*Figure 1a*). Specifically, participants were asked to count in their minds following a sequence of 50 ms pure tones presented at 250 ms onset-to-onset intervals, and they were instructed to count from 1 to 10 in groups of 5 numbers (**1**, 2, 3, 4, 5, **2**, 2, 3, 4, 5, **3**, 2, 3, 4, 5…**10**, 2, 3, 4, 5) until the 50 pure tones terminated. As a result, when the subjects were performing the task, the spectral peaks were significant at the

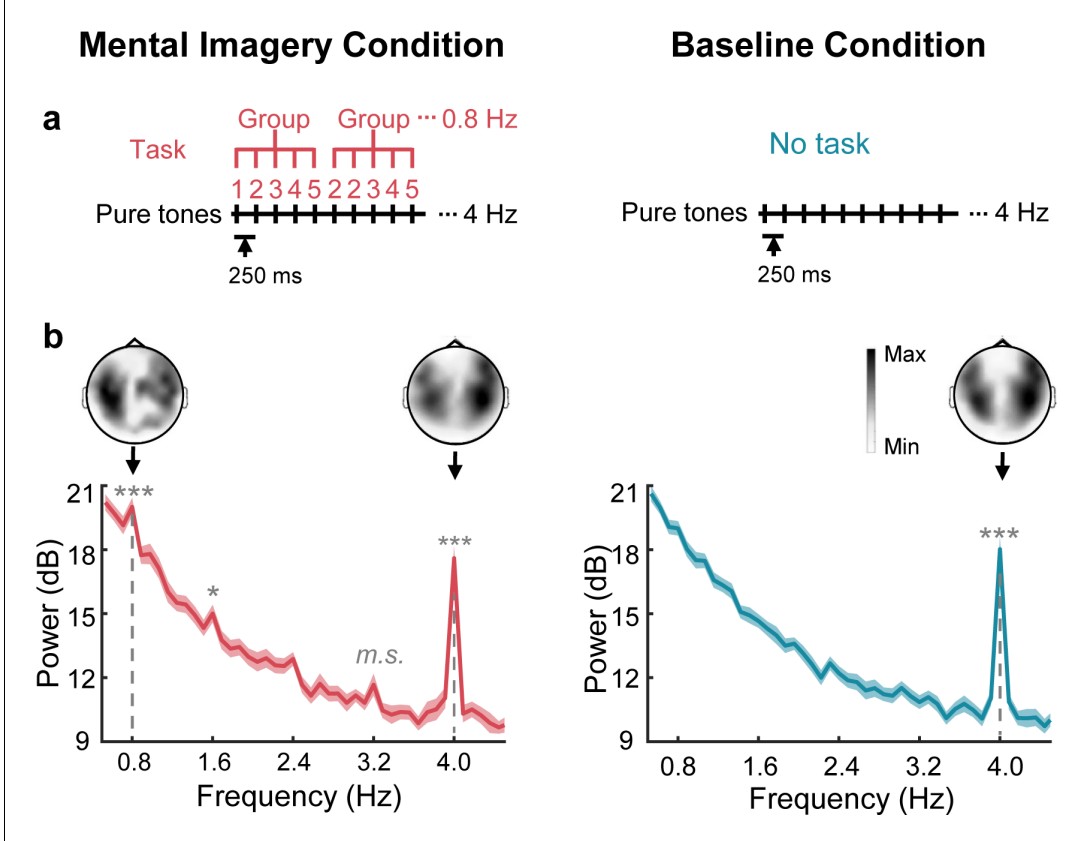

**Figure 1.** Sensor-level responses in neural tracking of speech mental imagery. (**a**) Under the mental imagery condition (left), a sequence of pure tones was presented with 250 ms onset-to-onset intervals. The participants were asked to loudly count in their minds by grouping every five numbers (with the imagery task), forming an imagery-rate frequency of 0.8 Hz and a stimulus-rate frequency of 4 Hz. Under the baseline condition (right), the participants passively listened to the same sounds without performing a task. (**b**) Significant spectral peaks were found at 0.8 Hz and its harmonics under the mental imagery condition, and significant 4 Hz peaks were found under both conditions (paired one-sided *t* test, FDR corrected). The topographic distribution was right-lateralized at 4 Hz but not at 0.8 Hz. The shaded area represents the standard error (SE). ***p<0.001, *p<0.05, *m.s.* represents marginally significant.

DOI: https://doi.org/10.7554/eLife.48971.002

The following source data is available for figure 1:

**Source data 1.** MEG sensor-level responses data.

DOI: https://doi.org/10.7554/eLife.48971.003

frequency of 0.8 Hz, corresponding to the grouping-rate rhythm of the mental imagery ($t_{19}$ = 5.33, p=4.2 × $10^{-4}$, Cohen's d = 1.19, one-tailed paired *t* test, false discovery rate [FDR] corrected), and at the frequency of 4 Hz, corresponding to the rate of the auditory stimuli ($t_{19}$ = 18.45, corrected p=3.0 × $10^{-12}$, Cohen's d = 4.13) (*Figure 1b*). An additional harmonic peak at 1.6 Hz was found at the significance level of 0.05 ($t_{19}$ = 3.18, corrected p=0.04, Cohen's d = 0.96). No significant peaks were observed at the harmonic frequencies of 2.4 Hz ($t_{19}$ = 2.46, corrected p=0.10, Cohen's d = 0.55) and 3.2 Hz ($t_{19}$ = 2.75, corrected p=0.07, marginally significant, Cohen's d = 0.62). In contrast, under the control (baseline) condition, in which the participant received identical auditory stimuli without the imagery task, a significant peak was observed only at the stimulus-rate frequency of 4 Hz ($t_{19}$ = 14.23, corrected p=3.1 × $10^{-10}$, Cohen's d = 3.18). The topographic distribution of the grand averaged power at the imagery-rate frequency (0.8 Hz) and stimulus-rate frequency (4 Hz) displayed bilateral responses. Specifically, the two-tailed paired *t* test showed that the averaged power over the sensors in the right hemisphere was significantly stronger than that in the left hemisphere at 4 Hz under both the mental imagery condition ($t_{19}$ = 3.78, p=0.001, Cohen's d = 0.84) and the baseline condition ($t_{19}$ = 3.21, p=0.004, Cohen's d = 0.72) but not at the imagery-rate frequency of 0.8 Hz under the mental imagery condition ($t_{19}$ = 0.34, p=0.72, Cohen's d = 0.08).

The comparison of the peak power between the two conditions confirmed the presence of imagery-induced cortical manifestations at the frequency of 0.8 Hz (*Figure 2a*). The peak power at the frequencies of 0.8 Hz and 4 Hz was calculated by subtracting the average of two neighboring frequency bins from the power of the peak frequency bin. A two-tailed paired $t$ test showed that the peak power at the imagery-rate frequency of 0.8 Hz when the participants induced rhythmic mental imagery was significantly larger than that under the baseline condition ($t_{19}$ = 2.40, p=0.027, Cohen's d = 0.54). No significant difference was observed between the two conditions at 4 Hz ($t_{19}$ = −0.30, p=0.764, Cohen's d = 0.07). Interestingly, the power of the neural responses at an imagery rate of 0.8 Hz was negatively correlated with that at a stimulus rate of 4 Hz when the participants induced the rhythmic mental imagery following the sound sequence ($r$ = −0.68, p=0.001) but not when they passively listened to the auditory stimuli ($r$ = −0.11, p=0.63) (*Figure 2b*).

## Brain regions activated by speech mental imagery

Using the frequency-tagging method, we identified the imagery-induced MEG manifestation in the frequency domain, but which neural networks generated such neural activity? Taking advantage of the previously developed minimum L1-norm source estimation methods (*Huang et al., 2006*; *Huang et al., 2012*; *Sheng et al., 2019*), we addressed this question by evaluating the neural sources in the MEG data within the single-frequency bins at the target frequencies of 0.8 Hz and 4 Hz. As a result, we found the following significant clusters activated at the imagery-rate frequency of 0.8 Hz (*Figure 3*, *Table 1*): (1) the left central-frontal lobe, including the pars opercularis part of the inferior frontal gyrus (operIFG), the inferior part of the precentral gyrus (PrG) and the postcentral gyrus (PoG), (2) the left inferior occipital gyrus (IOG) and (3) the right inferior parietal lobe (IPL), including the supramarginal gyrus (SMG) and the inferior part of the PoG (cluster level p<0.01, family-wise error [FWE] corrected with a voxel-level threshold of p<0.001). Regarding the stimulus-rate frequency of 4 Hz, a more extensive brain network involving the bilateral Heschl's gyrus (HG), right middle temporal gyrus (MTG), left SMG, bilateral superior temporal gyrus (STG), bilateral PrG, bilateral PoG and bilateral insula (INS) was significantly activated under both the mental imagery and the baseline conditions. An additional region in the right middle frontal gyrus (MFG) was also activated at 4 Hz under the mental imagery condition.

Then, we compared the activation of regions of interest (ROIs) in different brain regions at the following three significant spectral peaks: (1) at 0.8 Hz under the mental imagery condition, (2) at 4 Hz under the mental imagery condition and (3) at 4 Hz under the baseline condition (*Figure 4*). For

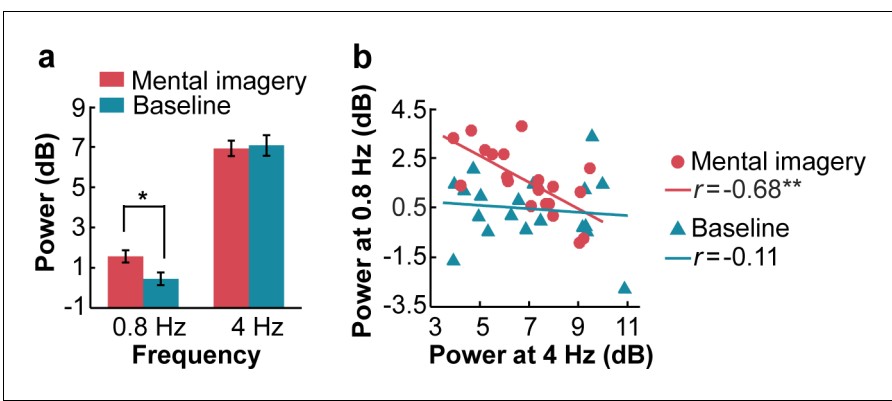

**Figure 2.** Power of spectral peaks at the imagery-rate frequency and stimulus-rate frequency. (a) Mental imagery increased the peak power at the imagery-rate frequency of 0.8 Hz. No significant difference in peak power at 4 Hz was observed between the conditions. (b) A significant negative correlation was observed between the grand averaged power at 4 Hz and 0.8 Hz under the mental imagery condition but not under the baseline condition. **p<0.01, *p<0.05.

DOI: https://doi.org/10.7554/eLife.48971.004

The following source data is available for figure 2:

**Source data 1.** Data for the power of spectral peaks.
DOI: https://doi.org/10.7554/eLife.48971.005

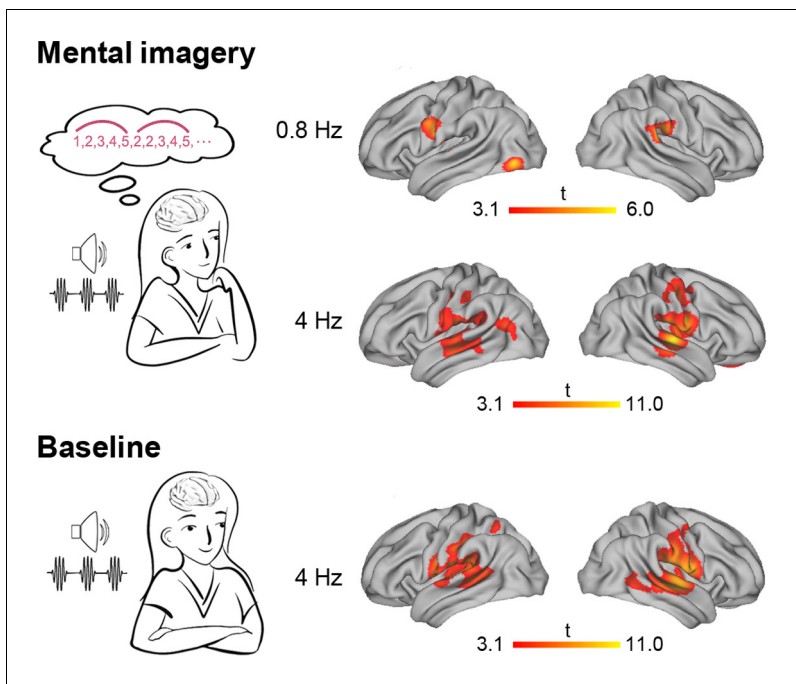

**Figure 3.** Source-level brain activation at the imagery-rate frequency and stimulus-rate frequency. Under the mental imagery condition (upper panel), the left central-frontal lobe, the left inferior occipital gyrus and the right inferior parietal lobe were activated at an imagery-rate frequency of 0.8 Hz, and a distributive brain network extending from the auditory cortex to the frontal and parietal regions was activated at a stimulus-rate frequency of 4 Hz (cluster level p<0.01, FWE corrected with a voxel-level threshold of p<0.001). Under the baseline condition (lower panel), stimulus-induced brain activation centring in the auditory temporal cortex was observed at 4 Hz.
DOI: https://doi.org/10.7554/eLife.48971.007

each brain region, the ROI centered at the peak voxel with the largest *t* value among the three levels was identified, and the amplitude of the activation was extracted from all 20 subjects. Two-way repeated measures ANOVAs (condition × brain region) for ROI activation showed a significant main effect of condition ($F_{2, 38}$ = 23.56, p<0.001, $\eta p^2$ = 0.55), a significant main effect of brain region ($F_{5.23, 99.41}$ = 6.66, p<0.001, $\eta p^2$ = 0.26), and a significant interaction between the two variables ($F_{8.09, 153.67}$ = 5.60, p<0.001, $\eta p^2$ = 0.23). To untangle the interaction, we performed post hoc comparisons with Bonferroni correction for each brain region. The results showed that in the left hemisphere, the neural responses in the left IOG at 0.8 Hz under the mental imagery condition were significantly stronger than those at 4 Hz under the mental imagery condition and at 4 Hz under the baseline condition (both p<0.05). In addition, the neural activation at 4 Hz under the imagery and baseline conditions was stronger than that at 0.8 Hz under the imagery condition in the left HG (p=0.058 and p<0.001, respectively), the left INS (p=0.012 and p<0.001, respectively) and the left SMG (p=0.032 and p<0.001, respectively). In the left STG, the activation amplitude at 4 Hz under the baseline condition was larger than that at 0.8 Hz under the imagery condition (p=0.006). In the right hemisphere, the activation observed at 4 Hz under the imagery and baseline conditions was stronger than that observed at 0.8 Hz under the imagery condition in most regions (right HG, both p<0.001; right STG, both p<0.001; right INS, both p=0.003; right PrG, p=0.029 and 0.016, respectively; right PoG, p=0.025 and 0.010, respectively; right MTG, p=0.010 and 0.005, respectively; all with Bonferroni correction), reflecting robust auditory neural representations of the 4 Hz sequence of sound stimuli. No significant difference was observed in the brain activation levels at 4 Hz between the imagery condition and baseline condition, except in the right MFG, in which the activation at 4 Hz during mental imagery was stronger than that at baseline (p=0.014). These data indicate that the isolated neural networks underlying speech mental imagery involve the left central-frontal lobe, the left inferior occipital lobe and the right inferior parietal lobule, and that these networks are disassociated from the sensory-driven processes.

**Table 1.** Major brain activity at the source level.

| Brain region | Peak MNI coordinate | | | T value |
|---|---|---|---|---|
| | X | Y | Z | |
| **Imagery condition at 0.8 Hz** | | | | |
| R postcentral gyrus | 56 | −22 | 29 | 4.0 |
| L postcentral gyrus | −51 | -7 | 23 | 4.9 |
| L precentral gyrus | −50 | -6 | 21 | 4.7 |
| L inferior frontal gyrus | −46 | 3 | 28 | 3.2 |
| L inferior occipital gyrus | −40 | −75 | -6 | 5.3 |
| R supramarginal gyrus | 60 | −31 | 24 | 6.0 |
| **Imagery condition at 4 Hz** | | | | |
| R superior temporal gyrus | 61 | −12 | 8 | 10.9 |
| L superior temporal gyrus | −51 | -5 | 5 | 5.7 |
| R middle temporal gyrus | 67 | −19 | -3 | 8.5 |
| L Heschl's gyrus | −50 | -7 | 5 | 5.3 |
| R Heschl's gyrus | 58 | −10 | 8 | 10.3 |
| R postcentral gyrus | 59 | −16 | 17 | 8.2 |
| L postcentral gyrus | −57 | −19 | 17 | 6.4 |
| R precentral gyrus | 55 | 0 | 19 | 7.3 |
| L precentral gyrus | −50 | -6 | 21 | 4.7 |
| R insula | 46 | −10 | 8 | 6.9 |
| L insula | −46 | -9 | 2 | 3.8 |
| R middle frontal gyrus | 6 | 50 | −11 | 4.0 |
| L supramarginal gyrus | −58 | −21 | 17 | 6.2 |
| **Baseline condition at 4 Hz** | | | | |
| R superior temporal gyrus | 56 | −16 | 1 | 11.0 |
| L superior temporal gyrus | −50 | −18 | 12 | 7.3 |
| R middle temporal gyrus | 55 | −27 | 0 | 7.3 |
| L Heschl's gyrus | −42 | −22 | 12 | 7.0 |
| R Heschl's gyrus | 54 | −13 | 8 | 10.8 |
| R postcentral gyrus | 61 | −15 | 15 | 8.8 |
| L postcentral gyrus | −53 | −11 | 18 | 8.7 |
| R precentral gyrus | 45 | -8 | 29 | 6.3 |
| L precentral gyrus | −54 | -4 | 21 | 5.9 |
| R insula | 48 | −10 | 4 | 6.9 |
| L insula | −38 | −24 | 22 | 6.8 |
| L supramarginal gyrus | −43 | −27 | 23 | 8.3 |

DOI: https://doi.org/10.7554/eLife.48971.006

## Localization of neural tracking speech imagery using sEEG

To further confirm the localization of the neural sources in the MEG data, we recorded sEEG responses to the same auditory stimuli with and without the imagery task. The sEEG responses are neurophysiological signals with a high temporal and spatial resolution recorded from patients with intracranial electrode implantation for the clinical assessment of epilepsy. We analyzed the power of sEEG data in the high gamma band (60–100 Hz), which is highly correlated with the neural firing rate (*Mukamel et al., 2011*; *Ray and Maunsell, 2011*). We found responsive contacts in five subjects with significant 4 Hz spectral peaks in the left posterior INS, left anterior SMG, bilateral middle STG, left HG and right middle MTG under the mental imagery and/or baseline conditions. Most

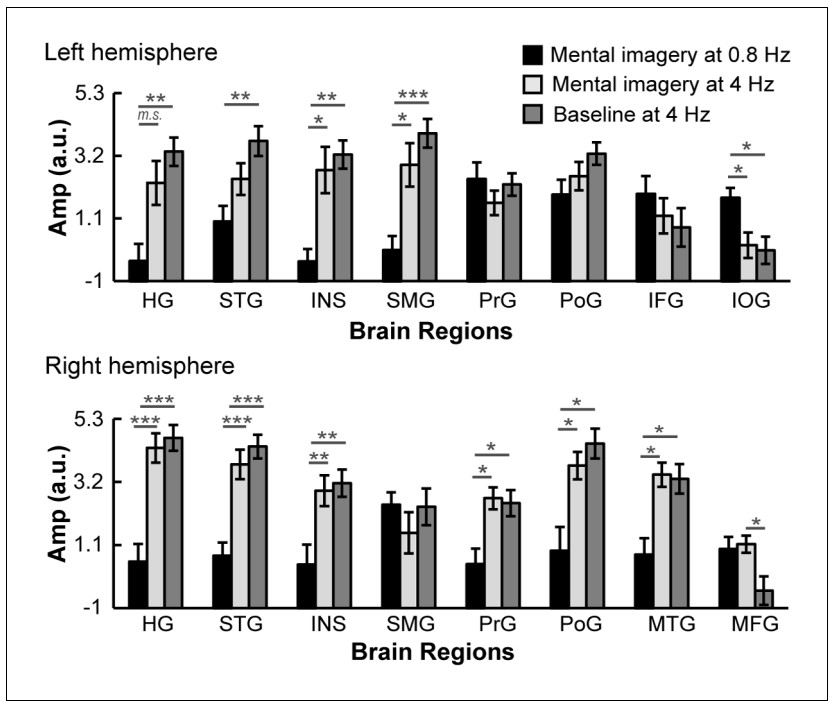

**Figure 4.** Comparisons of ROI activation among three significant spectral peaks. The brain activation in the left IOG at 0.8 Hz under the mental imagery condition was stronger than that at 4 Hz. The brain activation in the right MFG at 4 Hz under the mental imagery condition was stronger than that under the baseline condition. In the left HG, left STG, left INS, left SMG, right HG, right STG, right INS, right PrG, right PoG and right MTG, the activation at 4 Hz under the mental imagery condition and/or the baseline condition was stronger than that at 0.8 Hz under the mental imagery condition. Multiple comparisons were Bonferroni corrected. \*\*\*p<0.001, \*\*p<0.01, \*p<0.05, *m. s.* represents marginally significant.

DOI: https://doi.org/10.7554/eLife.48971.008

The following source data is available for figure 4:

**Source data 1.** ROI activation data.

DOI: https://doi.org/10.7554/eLife.48971.009

importantly, we found significant spectral responses at the imagery-rate frequency of 0.8 Hz when the participant was asked to perform a mental imagery task following the 4 Hz sound sequence ($t_{19}$ = 3.61, p=0.021, Cohen's d = 0.81, paired one-sided *t* test, FDR corrected) (*Figure 5a*). The imagery-responsive contact was localized in the operIFG, which is consistent with the MEG source results in the Montreal Neurological Institute (MNI) space (*Figure 5b*). Unlike in the baseline condition, under the mental imagery condition, the averaged time course of the high gamma activity in the operIFG periodically changed every 1.25 s (0.8 Hz), with an increasing amplitude near the last number of each mentally constructed group (*Figure 5c*). Due to limited intracranial cases, we did not obtain sEEG data from the regions of the left IOG and right IPL, which were the other two neural clusters in the MEG source results.

In addition, we found significant spectral responses at the stimulus-rate frequency of 4 Hz (*Figures 5a, d and e* for subjects 1, 2, and 3; *Figure 5—figure supplement 1* for subjects 4 and 5; *Table 2* for statistical results). In the left hemisphere, significant 4 Hz peaks were observed at the contacts localized in the posterior INS, anterior SMG and HG under the mental imagery condition and at the contacts localized in the middle STG, posterior INS, anterior SMG and HG under the baseline condition. In the right hemisphere, we found significant spectral responses at 4 Hz in the middle STG and middle MTG. Specifically, the right middle STG was responsive to the 4 Hz stimuli under the baseline condition and the mental imagery condition. The right middle MTG was responsive to the 4 Hz stimuli only under the baseline condition. These brain regions with significant

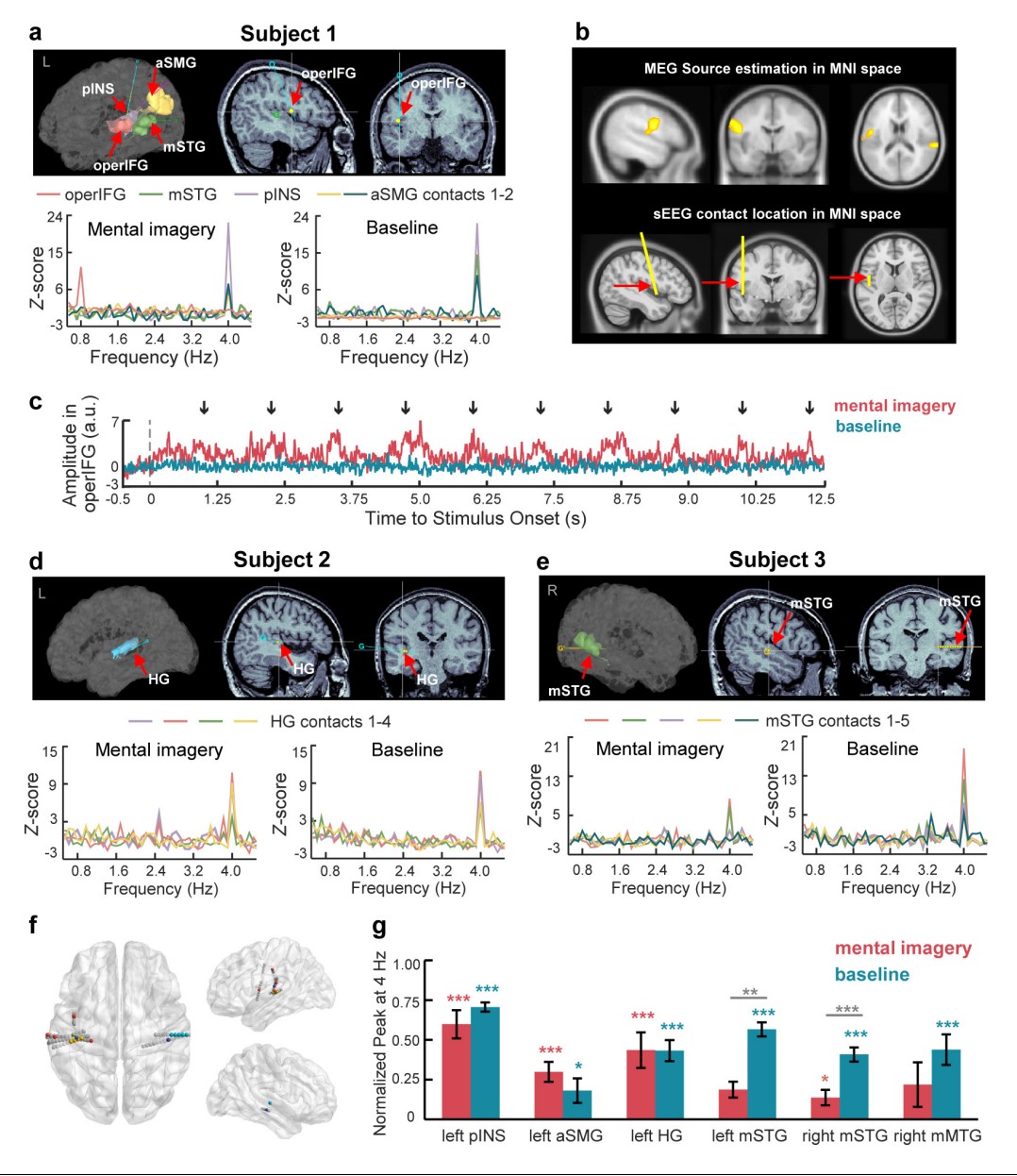

**Figure 5.** SEEG responses to speech mental imagery. (a) Subject one with contacts in the left operIFG, middle STG, posterior INS and anterior SMG. Significant spectral responses at 0.8 Hz were detected in the operIFG, and the remaining contacts were responsive to the 4 Hz frequency. (b) The cluster including the operIFG from MEG sources is presented in the upper panel. The electrode with contact in the operIFG in Subject one was projected in the MNI space and is shown in the lower panel. SEEG localization supports the MEG source estimation results. (c) Averaged time course of high gamma activity in the operIFG in Subject 1. Periodical changes presented every 1.25 s (0.8 Hz) with increasing amplitude near the onset of the last number (black arrow) in each mentally constructed group under the mental imagery condition (red line) but not the baseline condition (blue line). (d) Subject two with contacts in the left HG. Significant spectral peaks were observed at 4 Hz in the HG. (e) Subject three with contacts in the right middle STG. Significant spectral peaks were observed at 4 Hz in the middle STG. (f) Contact locations in the MNI space. Different colored marks represent each subject's responsive contacts with significant spectral peaks at 0.8 Hz and/or 4 Hz. Gray marks represent non-responsive contacts with no significant spectral peaks. (g) In each brain region with contacts responsive at 4 Hz, the normalized peaks were significantly larger than zero under the baseline condition (blue stars). Among these peaks, the normalized peaks were significant in the left posterior INS, left anterior SMG, left HG and right middle STG (red stars). The peaks in the

*Figure 5 continued on next page*

*Figure 5 continued*

bilateral middle STG were smaller under the mental imagery condition than the baseline condition (gray stars). ***p<0.001, **p<0.01, *p<0.05.

DOI: https://doi.org/10.7554/eLife.48971.010

The following source data and figure supplement are available for figure 5:

**Source data 1.** sEEG response data.
DOI: https://doi.org/10.7554/eLife.48971.012
**Figure supplement 1.** SEEG results of subjects 4–5.
DOI: https://doi.org/10.7554/eLife.48971.011

spectral peaks at 4 Hz mainly agreed with the estimated neural sources based on the MEG data (*Table 1*).

Subsequently, we combined all the trials at the contacts localized in the same brain regions across different subjects. *Figure 5f* displays the localization of contacts in the MNI space from all five subjects. We calculated the normalized peak power in each brain region and examined the following two questions: (1) whether the 4 Hz peak responses were significant and (2) whether there were significant differences between the conditions (*Figure 5g*). The one-tailed paired $t$ tests revealed significant 4 Hz peak powers under the baseline condition in several brain areas, including the left posterior INS ($t_{19}$ = 24.82, p<0.001, Cohen's d = 3.05), left anterior SMG ($t_{58}$ = 2.32, p=0.012, Cohen's d = 0.30), left HG ($t_{99}$ = 10.10, p<0.001, Cohen's d = 1.07), bilateral middle STG (left,

**Table 2.** Statistical details for the sEEG results

| Contact localization | Subject no. | Df | T | Corrected P | Cohen's d |
|---|---|---|---|---|---|
| **Mental imagery condition at 4 Hz** | | | | | |
| L posterior insula | 1 | 19 | 5.57 | <0.001 | 1.24 |
| L anterior supramarginal gyrus | 1 | 19 | 3.40 | 0.034 | 0.76 |
| | 1 | 19 | 2.99 | 0.084 | 0.67 |
| | 5 | 18 | 3.29 | 0.046 | 0.75 |
| L Heschl's gyrus | 2 | 15 | 5.00 | 0.002 | 1.25 |
| | 2 | 15 | 4.64 | 0.004 | 1.16 |
| | 4 | 19 | 4.57 | 0.002 | 1.02 |
| R middle superior temporal gyrus | 3 | 19 | 4.08 | 0.007 | 0.91 |
| | 3 | 19 | 5.29 | <0.001 | 1.18 |
| **Baseline condition at 4 Hz** | | | | | |
| L middle superior temporal gyrus | 1 | 19 | 4.47 | 0.003 | 1.00 |
| L posterior insula | 1 | 19 | 8.06 | <0.001 | 1.80 |
| L anterior supramarginal gyrus | 1 | 19 | 3.35 | 0.037 | 0.75 |
| | 5 | 18 | 3.43 | 0.034 | 0.79 |
| L Heschl's gyrus | 2 | 19 | 4.41 | 0.003 | 0.99 |
| | 2 | 19 | 4.58 | 0.002 | 1.02 |
| | 2 | 19 | 4.60 | 0.002 | 1.03 |
| | 2 | 19 | 3.44 | 0.031 | 0.77 |
| R middle superior temporal gyrus | 3 | 19 | 5.29 | <0.001 | 1.18 |
| | 3 | 19 | 4.35 | 0.004 | 0.97 |
| | 3 | 19 | 3.81 | 0.013 | 0.85 |
| | 3 | 19 | 3.97 | 0.009 | 0.89 |
| | 3 | 19 | 3.79 | 0.014 | 0.85 |
| R middle middle temporal gyrus | 5 | 18 | 4.52 | 0.003 | 1.34 |

DOI: https://doi.org/10.7554/eLife.48971.013

$t_{19}$ = 8.60, p<0.001, Cohen's d = 1.92; right, $t_{99}$ = 9.28, p<0.001, Cohen's d = 0.93) and right middle MTG ($t_{18}$ = 4.59, p<0.001, Cohen's d = 1.05). Among these regions, the 4 Hz peaks were also significantly larger than zero under the mental imagery condition in the brain areas of the left posterior INS ($t_{19}$ = 6.71, p<0.001, Cohen's d = 1.50), left anterior SMG ($t_{58}$ = 4.73, p<0.001, Cohen's d = 0.62), left HG ($t_{83}$ = 8.97, p<0.001, Cohen's d = 0.98) and right middle STG ($t_{99}$ = 2.69, p=0.004, Cohen's d = 0.27).

Moreover, the comparisons between the conditions showed that the power at 4 Hz peaks in the bilateral middle STG under the mental imagery condition was reduced compared to that under the baseline condition (left middle STG, $t_{38}$ = −2.93, p=0.006, Cohen's d = 0.93; right middle STG, $t_{198}$ = −4.05, p<0.001, Cohen's d = 0.57, two-tailed independent t test). Notably, the power at the 0.8 Hz peak in the operIFG during the imagery task was increased compared to that under the baseline condition ($t_{38}$ = 3.52, p=0.001, Cohen's d = 1.11). This result further supports the MEG results showing the reverse correlation between the power of the top-down neural activity induced by mental imagery and the power of the bottom-up stimulus-driven processes (*Figure 2b*).

## Discussion

Our data show that the neural activity induced by speech mental imagery can be tracked in the frequency domain and distinguished from stimulus-driven processing. The frequency-tagging method we applied in the imagery task casts light on the neural dynamics of the internally generated representations (i.e. speech mental imagery). More specifically, the left central-frontal lobe, the left inferior occipital lobe and the right inferior parietal lobe contribute to the spectral responses generated by speech mental imagery. Furthermore, the detection of imagery-induced neural signals in the left central-frontal lobe, particularly the region of the operIFG, in rare intracranial cases confirms the reliability of the minimum L1-norm estimation methods in solving the inverse problem for MEG data in our study.

Mental imagery has been proposed to be essentially an attention-guided memory retrieval process, for example, in the visual domain (*Ishai et al., 2002*). Mental imagery of speech not only recruits the pathway by which phonological, lexical, or semantic details are retrieved from the memory system but also requires planned articulation that is simulated internally to enable the individuals to have the subjective experiences of their mental imagery (*Tian and Poeppel, 2012*; *Tian et al., 2016*). In our study, the task of rhythmic inner counting produced cyclical neural signals that were captured in the frequency domain, reflecting the top-down regulation of the internally constructed mental imagery. The operational definition of speech mental imagery condition, namely, the rhythmic inner counting task, is not only related to the internal representations of speech but also requires high levels of working memory. To count following a rapid sequence of sounds presented at a 250 ms onset-to-onset interval, the brain must maintain the mental representations of the last group of 5 numbers (or at least of the beginning number) and construct the ongoing imagery of the subsequent group of 5 numbers concurrently with the sound stimuli. Additionally, the participants needed to verbally name the numbers and count 'loudly' in their minds. It is worth mentioning that the rhythmic inner counting task in our study eliminated the influence of bottom-up sensory-driven processes by using the same pure tones as the stimuli and investigated the pure top-down processes of speech mental imagery independent of bottom-up perception. This is different from the typical silent counting task used in evoking P300 responses, which largely depends on the sensory perception and anticipation of the bottom-up physical features of the stimulus. The 0.8 Hz rhythmic mental operation activated the region of the left IOG, which is reported to be associated with verbal representation (*Saito et al., 2012*) and memory maintenance in numeral processing (*Daitch et al., 2016*), and the region of the right IPL, which is associated with auditory working memory (*Paulesu et al., 1993*) and has also been proposed to be activated in imagined singing and imagined movement (*Kleber et al., 2007*; *Lebon et al., 2018*). In addition, our study shows that the ventral PrG and PoG are also activated during the construction of speech imagery, which is consistent with previous findings that stronger activation of the sensorimotor cortex induced by articulatory imagery than hearing imagery (*Tian et al., 2016*) and the direct neural signal from the sensorimotor cortex contributed to the decoding of imagined speech and silent reading (*Martin et al., 2014*; *Martin et al., 2016*). Nevertheless, a firm conclusion on the contribution of the sensorimotor cortex to the construction of speech mental imagery will only be possible after controlling for any articulator movement in future

studies. Furthermore, consistent with previous findings in fMRI studies suggesting that Broca's area (operIFG) is activated during the silent articulation of speech (*Aleman et al., 2005*; *Kleber et al., 2007*; *Papoutsi et al., 2009*; *Koelsch et al., 2009*; *Price, 2012*; *Tian et al., 2016*), the imagery-enhanced spectral responses in the region of the operIFG are not only identified in the MEG neural sources but also detected in the intracranial focal recordings, confirming the involvement of the higher order frontal cortex in the dynamic construction of speech mental imagery. Taken together, these results show for the first time that the disassociated neural networks underlying the internal organization of speech mental imagery are independent of auditory perception. Because we identified only the operIFG as a neural source in the speech mental imagery due to the limited intracranial cases in the present sEEG experiment, more work is still needed to investigate the role of the left IOG and right IPL in the neural pathways underlying speech mental imagery.

Mixed evidence exists regarding the relationship between imagery and perception. One line of evidence suggests that mental imagery and sensory perception share common processes; for instance, the visual cortex can also be activated by visual imagery (*Kosslyn et al., 1999*; *Stokes et al., 2009*; *Dijkstra et al., 2017*), and the auditory cortex is recruited during the imagination of music (*Zatorre and Halpern, 2005*; *Kraemer et al., 2005*). However, other studies have suggested that differentiated neural substrates underlie these two processes; for example, brain-damaged patients with impaired object recognition have normal visual imagery (*Behrmann et al., 1992*), and the primary auditory cortex is activated by overt singing but not imagined singing (*Kleber et al., 2007*). *Tian et al. (2018)* shed light on this issue by investigating the imagery-perception adaptation effect in speech mental imagery and found that imagined speech modulates the perceived loudness of a subsequent speech stimulus, demonstrating the overlapping neural underpinnings of the auditory cortex underlying the two processes of imagery and perception. Limited by the imagery-perception repetition paradigm, Tian et al. did not illustrate the role of non-overlapping brain regions that might be involved in the modulatory effect of imagery on perceived loudness. In our study, we address this gap by disassociating the two processes of imagery and perception, which were precisely frequency-tagged, and identifying the specific neural sources of speech mental imagery independent of auditory perception. We show that the auditory cortex was not engaged in neural activation at an imagery-rate frequency of 0.8 Hz. Instead, some isolated neural clusters, including the higher order frontal cortex, were recruited at the imagery-rate frequency, reflecting the non-overlapping neural populations underlying the two distinctive processes of speech imagery and auditory perception. Notably, during the imagery task, the spectral power at the stimulus-rate frequency was weakened at the local contacts of the middle STG according to the sEEG recordings, indicating a potential imagery-driven suppression of auditory perception that fits well with the results of previous adaptation studies (*Tian and Poeppel, 2013*; *Tian and Poeppel, 2015*; *Ylinen et al., 2015*; *Whitford et al., 2017*; *Tian et al., 2018*). In addition, we uncovered a reverse correlation between the grand averaged power at the imagery-rate frequency and the stimulus-rate frequency (*Figure 2b*), which might have resulted from the reallocation of limited cognitive resources between the low-level auditory representation and high-level imagery construction. The crucial role of the (left-lateralized) frontal cortex in the top-down control mechanism has been well demonstrated in the speech perception literature with the account of predictive coding (*Park et al., 2015*; *Cope et al., 2017*). Interestingly, *Sohoglu et al. (2012)* also found opposite effects of top-down expectations and bottom-up sensory details on the evoked response in the STG. Based on these findings, we propose that the higher order cortex (i.e. the inferior frontal cortex) is engaged in the dynamic organization of speech mental imagery, which might modulate neural activity in the lower-order auditory cortex. Determining the functional connectivity between the higher-order cortex and auditory cortex during the construction of speech mental imagery may be an important direction in future studies.

In our study, the same auditory stimuli (a 4 Hz sequence of pure tones) were played under both the mental imagery condition and baseline condition. As a result, strong spectral peaks with similar power amplitudes (*Figure 2a*) and right-lateralized topographic distributions (*Figure 1b*) at the sensor level were observed at a frequency of 4 Hz, reflecting robust stimulus-driven auditory responses. Therefore, we mainly refer to 4 Hz as the 'stimulus-rate' frequency, but we should be aware that the mental imagery of each number during the counting task was first produced at 4 Hz and then internally organized into groups at 0.8 Hz. When comparing the 4 Hz responses between conditions at the source level, we found greater activation of the right MFG in the mental imagery condition

(*Figure 4*), which might reflect the additive effect of generating the speech mental imagery of each number during the counting task. Previous fMRI findings have suggested that the MFG is activated by the auditory imagery of music (*Zvyagintsev et al., 2013*; *Lima et al., 2015*) and can be activated by hearing imagery (HI) of speech compared to articulatory imagery (AI) (*Tian et al., 2016*), but the results of our MEG experiment showed that the MFG also contributes to the processing of articulation imagery. From the perspective of inner speech theories (*Alderson-Day and Fernyhough, 2015*), HI, which refers to the inner representation of another's voice, is distinct from articulatory imagery, which refers to covert speech production, including motor preparation and verbal working memory maintenance (*Acheson et al., 2011*; *Tian and Poeppel, 2013*; *Tian et al., 2016*). To ensure that the imagery task appeared to be natural and reflected subjects' daily experience, we set the imagery task in our study as the mental operation of counting silently, which can be treated as a type of articulatory imagery, without asking the participants to strictly distinguish between HI and articulatory imagery. Therefore, care should be taken when comparing our results with other neuroimaging findings in the research area of HI of speech or auditory imagery of music. In the future, whether the neural mechanism of speech mental imagery uncovered in our paradigm can be extended to a richer account of inner speech must be considered.

Note that the auditory cortex was not significantly activated at the imagery-rate frequency. This result is different from the previous finding that similar neural responses were observed for overt and covert hearing tasks (*Tian and Poeppel, 2010*), which can be explained from two aspects. Firstly, the neural pathways underlying speech mental imagery are strongly modulated by the specific imagery requirement (i.e. articulatory imagery or HI) and task demand (*Tian and Poeppel, 2010*; *Tian and Poeppel, 2012*; *Tian and Poeppel, 2013*; *Tian et al., 2016*; *Tian et al., 2018*). Unlike the detailed requirements for imagery used in *Tian and Poeppel (2010)*, in our study, we instructed the participants to count in their minds without asking them to strictly distinguish between articulatory imagery and HI. Secondly, and most importantly, we for the first time applied frequency-tagging paradigm and disassociated the neural activity induced by rhythmic inner counting from that elicited by bottom-up auditory perception. The activation of auditory cortex was tagged and detected at the stimulus-rate frequency instead of imagery-rate frequency. These differences in the task requirements and experimental paradigm might have led to the different findings concerning the involvement of the auditory cortex (*Tian and Poeppel, 2010*) in forming speech mental imagery.

Due to the absence of overt behavioral outcomes in mental imagery tasks, it is difficult or impossible to directly correlate the neural signals to the corresponding behavioral performance during the imagery task. To overcome this obstacle, some behavioral measurements have been developed in previous imagery studies, such as participant self-report assessments and judgement tasks regarding the content and nature of inner speech (*Hurlburt et al., 2013*; *Hurlburt and Heavey, 2015*; *Lima et al., 2015*), but these approaches still largely rely on subjective experience without objective verification. Our study did not obtain behavioral data from the participants during the task but controlled for their behavioral performance via a preceding training session in which the participants were asked to report their performance of inner number generation and ensured that after training, they were able to perform the imagery task correctly. In addition, in the sEEG experiment, we recorded the participants' subjective reports to exclude the incorrect trials from the analysis. A key issue that should be further addressed is advancing the paradigm to study speech imagery using behavioral measurements. Recently, an enlightening study conducted by *Martin et al. (2018)* proposed a novel task that recorded both the output of a keyboard (without producing real sounds) and the neural activity when a musician imagined the music he or she was playing. Thus, the objective data from the recorded sound of the keyboard were simultaneously obtained and could be analyzed with the neural representations of the music imagery. In addition, intracranial neural signals from the auditory cortex and sensorimotor cortex can be used to reconstruct perceived and imagined speech (*Martin et al., 2014*; *Akbari et al., 2019*), providing a new account for decoding the neural signals induced by speech imagery. Inspired by these studies, more work needs to be performed to examine the correlations between possible behavioral manifestations and the neural substrates underlying mental imagery to deepen the understanding of speech mental imagery and advance its application in brain-machine interfaces.

In summary, our study applied a frequency-tagging paradigm to track the neural processing time scales in line with the generation of speech mental imagery and the stimulus-driven activity using both the neuronal measures (MEG) and direct recordings from the cortex (sEEG). We uncovered a

distinctive neural network underlying the dynamic construction of speech mental imagery that was disentangled from sensory-driven auditory perception.

## Materials and methods

### Participants
Twenty Mandarin-speaking young students (8 females and 12 males, 24.6 ± 3.6 years old, all right-handed) participated in the MEG experiment. The participants all had normal hearing abilities and reported having no history of mental disorders. Written informed consent was obtained from all participants. The participants were paid, and the experimental protocol was approved by the Peking University Institutional Review Board.

### Stimuli
The auditory signal was a 50 ms pure tone generated by Adobe Audition software (Adobe Systems Incorporated, San Jose, CA). The frequency of the pure tone was 440 Hz, and the sampling rate was 1.6 kHz.

### Procedures
During the experiment, each participant was seated in a dimly lit and magnetically shielded room (VACUUMSCHMELZE GmbH and Co. KG, Hanau, Germany) with a distance of 1 m from a projected screen. The acoustic signals were delivered to the participants using a MEG-compatible inserted ear-phone. Two blocks (a mental imagery block and a baseline block) were performed, and each block included 15 trials. The presentation order of the blocks was counterbalanced across subjects. Within each trial, the instruction was first presented on the screen. For the imagery block, the instructions were "Please count loudly in mind with five numbers in a group, **1**, 2, 3, 4, 5, **2**, 2, 3, 4, 5, **3**, 2, 3, 4, 5…**10**, 2, 3, 4, 5, following the rhythm of the sound until the sound is terminated'. In this counting task, the first number in each group increased from '1' to '10', and the remaining numbers in the group were always '2, 3, 4, 5'. For the baseline block, the instructions were 'Please listen carefully to the sound until the sound is terminated'. The participants pressed the button when they were ready, and then a central fixation cross remained on the screen until the trial terminated. After a silent random interval ranging from 1 to 1.5 s, a pure tone was presented 50 times repeatedly with an onset-to-onset interval of 250 ms. Thus, the frequency of the presentation of the pure tones was 1/0.25 Hz, that is 4 Hz. More importantly, the frequency of the occurrence of the rhythmic imagery was 1/1.25 Hz, that is 0.8 Hz. The sound sequence in each trial lasted for 12.5 s.

Before the formal experiment, the subjects received a training session in which they were asked to orally report their inner performance after each trial. The training session was supervised to ensure that each subject could perform the task correctly. After the training, all the participants were able to generate the mental imagery of 50 numbers with every five numbers in a group following the 50 pure tones.

### MEG recording and structural MRI data acquisition
The continuous neuromagnetic signal was recorded using a 306-channel, whole-head MEG system (Elekta-Neuromag TRIUX, Helsinki, Finland) at Peking University. Before each block started, the head position of each subject was determined by four head-position indicator (HPI) coils. The electrooculogram (EOG) signal was simultaneously captured by two electrodes placed near the eyes as follows: one electrode was placed above the right eye, and one electrode was placed below the left eye. The continuous MEG data were on-line bandpass filtered at 0.1–330 Hz. The sampling rate was 1000 Hz.

The subjects' structural MRI images were obtained with a 3T GE Discovery MR750 MRI scanner (GE Healthcare, Milwaukee, WI, USA). A three-dimensional (3D) fast gradient-recalled acquisition in the steady state with a radiofrequency spoiling (FSPGR) sequence was used to obtain $1 \times 1 \times 1$ mm$^3$ resolution T1-weighted anatomical images. We co-registered the MEG data with the MRI data based on the location of three fiducial marks (the nasion and two pre-auricular points) and approximately 150 digitalization points on each subject's scalp.

## Data pre-processing

The raw data were first processed using the temporal Signal Space Separation (tSSS) method (*Taulu and Simola, 2006*) implanted in Maxfilter software (Elekta-Neuromag) for magnetic artefact suppression. Manually identified bad channels (3–5 channels per subject) with excessive noise were excluded from further analysis. The head position of the second block was co-registered to that of the first block using MaxMove software (Elekta-Neuromag). The independent component analysis (ICA) method was applied to remove the ocular artefacts using the MNE software (http://www.martinos.org/mne/). Specifically, the ICA component with the highest correlation to the EOG signal was selected and rejected. Then, a bandpass filter (0.2 to 60 Hz) and a notch filter (50 Hz power-line interference) were applied to the de-noised data using the FieldTrip toolbox (http://www.fieldtrip-toolbox.org), during which edge effects after filtering were eliminated by padding extra data on either side of each epoch. Finally, the event-related field (ERF) response was obtained by averaging the data with a time window of −0.5 to 12.5 s.

The cortical reconstruction and volume-based segmentation were completed based on each subject's T1-weighted image using Freesurfer (http://surfer.nmr.mgh.harvard.edu/). A realistic boundary element method (BEM) model was employed to solve the forward problem. A 5 mm cubic grid was prepared as the source space, resulting in approximately 10,000 nodes in the entire brain.

## Sensor-level analysis

The ERF response acquired during the pre-processing stage was modified by excluding the first 1.25 s of the stimulus to eliminate the transient response. Therefore, the stimulus length of the ERF was 11.25 s, leading to a frequency resolution of 1/11.25 Hz, that is 0.089 Hz. To pre-whiten all the magnetometers and gradiometers, a diagonal noise covariance matrix was calculated based on the 500 ms pre-stimulus baseline. Then, the pre-whitened data were transformed into the frequency domain using the fast Fourier transform (FFT). The average response power of all channels was calculated to compare the rhythmic response among the different conditions.

## Source estimation method

The present source estimation method was rooted in the previously developed minimum L1-norm estimation methods (*Huang et al., 2006*; *Huang et al., 2012*; *Sheng et al., 2019*), which offer high spatial resolution for MEG analysis. The $m \times t$ sensor waveform matrix, where $m$ and $t$ represent the number of MEG sensors and the number of time points, respectively, was transformed using the FFT into frequency domain matrix $\mathbf{K}$, which included both real and imaginary parts. To focus on the peak response at a given frequency (for example, 0.8 Hz in the imagery block during which the rhythm of mental imagery was manifested), we chose the single-frequency bin $\mathbf{K}_f$ for the source estimation, where $f$ refers to the target frequency bin. Furthermore, a convex second-order cone programming (SOCP) method for solving the minimum L1-norm problem was applied to minimize the bias toward the coordinate axes (*Ou et al., 2009*) as follows:

$$min \sum_{i=1}^{n} \boldsymbol{w}_i \sqrt{\left(\omega_{i,real}^{\theta}\right)^2 + \left(\omega_{i,imag}^{\theta}\right)^2 + \left(\omega_{i,real}^{\phi}\right)^2 + \left(\omega_{i,imag}^{\phi}\right)^2}$$

$$s.t. \mathbf{K}_f = \mathbf{G}\Omega_f \tag{1}$$

where $\boldsymbol{w}$ refers to the depth weighting vector, $\mathbf{G}$ refers to the $m \times 2N$ gain matrix obtained from the BEM calculation, $n$ is the number of the source grid, and $i$ is the index of the source grid. $\theta$ and $\phi$ refer to the two dominant directions of the gain matrix (i.e. dominant source directions) using singular value decomposition. The solution is the $n \times 1$ complex vector $\varsigma_f$.

## Statistical analysis

For the sensor-level analysis, a one-tailed paired $t$ test was conducted to evaluate whether a significant spectral peak occurred at a frequency bin compared to the average of two neighboring frequency bins. This test was applied to all the frequency bins between 0.5 and 4.5 Hz, and the p values were adjusted by a false discovery rate (FDR) correction for multiple comparisons.

For the source-level analysis, each subject's ERF response ranging from 1.25 to 12.5 s was first transformed into the frequency domain for the source estimation. The brain activation to the target frequency bin was obtained by calculating the root mean square (RMS) of the real and imaginary parts. A control state was created by averaging the sources of two neighbor frequency bins adjacent to the target frequency. Before the group analysis, individual brain result was projected to the MNI-152 template using FSL software (https://fsl.fmrib.ox.ac.uk/fsl/fslwiki/) and spatially smoothed with a Gaussian kernel of 5 mm full-width at half-maximum (FWHM). The brain results were further log-transformed to reduce the skewness of the distribution of the minimum L1-norm solutions (*Huang et al., 2016*). A brain activation map at a certain frequency bin under each condition was obtained by comparing the brain responses with the average of their neighboring controls using a one-tailed paired *t* test (cluster level p<0.01, family-wise error [FWE] corrected with a voxel-level threshold of p<0.001). Furthermore, several regions of interest (ROIs) were identified by finding the local maximum of the significant neural clusters, and a 5 mm cube was used to extract the activation data from each ROI. Then, the brain activation was quantitatively compared among the following three levels: (1) at the frequency of 0.8 Hz under the imagery condition; (2) at the frequency of 4 Hz under the imagery condition; and (3) at the frequency of 4 Hz under the baseline condition. Repeated measures ANOVAs were used to examine the differences among the three levels. A Greenhouse-Geisser correction was applied for violation of sphericity. Post hoc comparisons were conducted using Bonferroni corrections. Here, non-independent ROI selection criteria were chosen to quantitatively compare the source-level neural activation between conditions; thus, we interpreted the results with caution to avoid overestimation of the ROI data.

## SEEG experiment
### SEEG Participants
Five subjects (three females; 27.6 ± 12.0 years old, range 14–46 years old) who were undergoing clinical intracerebral evaluation for epilepsy participated in the sEEG experiment. All the participants were right-handed and had normal hearing abilities. Before the experiment, the participants gave their written consent for participation, including permission for scientific publication. As one subject was a minor, written informed consent from his/her parent was obtained. Intracerebral electrodes were stereotactically implanted within the participants' brains solely for clinical purposes. All analyses were performed offline and did not interfere with the clinical management of the subjects. The sEEG study was approved by the Ethics Committee of the Sanbo Brain Hospital, Capital Medical University.

### SEEG recordings
Each 0.8 mm diameter intracerebral electrode contained 12–18 independent recording contacts of 2 mm in length and 1.5 mm apart from each other. The reference site was attached to the skin of each subject's forehead. For each subject, the cortical reconstruction and segmentation were computed based on the T1-weighted image obtained before the electrode implantation using BrainSuite software (http://brainsuite.org/). Then, the CT with the electrode localizations and the anatomical images were co-registered and normalized to the MNI-152 template. The contact locations were identified and visualized with MNI coordinates using the Brainstorm toolbox (http://neuroimgage. usc.edu/brainstorm/) and the BrainNetViewer toolbox (https://www.nitrc.org/projects/bnv/) in the MATLAB environment. The intracranial neural responses were collected with the Nicolet clinical amplifier. The sampling rate was 512 Hz in four subjects (subjects 1 to 4) and 2048 Hz in one subject (subject 5).

### SEEG procedures
The subjects listened to the same stimuli and performed the same task as the healthy subjects in the previous MEG experiment. The participants were asked to count in their mind every five numbers in a group (imagery-rate frequency = 0.8 Hz) following a sequence of 50 pure tones (stimulus-rate frequency = 4 Hz). The auditory stimuli and trial structure were identical to those used in the MEG study, except that at the end of each trial under the mental imagery condition, the participants were instructed to orally report whether they had counted correctly following all the sounds. Then, the experimenter recorded their subjective report (correct/incorrect). Only correct trials were included in

the subsequent analyses. In addition, the number of trials under each condition was dependent on each patient's physical status, while a fixed number of trials was used in the MEG experiment. Furthermore, to reduce the confusion caused by the imagery task among the patients, the baseline condition was presented first, followed by the mental imagery condition. Before the formal mental imagery trials, the participants engaged in a practice session to ensure that they had fully understood the imagery task.

## SEEG data processing

The sEEG data from each contact were first analyzed separately, and the high gamma activity, which is related to the neural firing rate (*Mukamel et al., 2011*; *Ray and Maunsell, 2011*), was extracted by a bandpass filter of 60–100 Hz. The instantaneous amplitude envelope was obtained from the filtered data by applying the Hilbert transform and then converted to the frequency domain in each trial using the FFT. A significant spectrum peak was reported if the response power at the target frequency was significantly stronger than the average of its two neighboring frequencies using a one-tailed paired *t* test among all the trials under the same condition. This test was applied to all frequency bins between 0.5 and 4.5 Hz, and the p values were FDR corrected for multiple comparisons. The spectral power at each contact under both conditions was transformed to the *Z*-score for visual presentation (*Figure 5a,c,d* and *Figure 5—figure supplement 1*). Specifically, the *Z*-score was computed as the difference between the power response in a frequency bin and the mean power response between 0.5 Hz and 4.5 Hz excluding the frequency bins of 0.8 Hz and 4 Hz divided by the standard deviation (SD) of the power response in these frequency bins.

To compare the responses among the contacts, the spectral peaks at each contact were first extracted and normalized as follows:

$$Normalized\ Peak = \frac{Power_{target} - Power_{neighboring}}{Power_{target} + Power_{neighboring}} \tag{2}$$

where $Power_{target}$ refers to the response power at a target frequency (i.e. 4 Hz), and $Power_{neighboring}$ refers to the averaged response power at the two frequency bins neighboring the target frequency. For each brain region, a one-tailed paired *t* test was used to evaluate whether the normalized peak of all trials in that region was significantly larger than zero. Then, a two-tailed independent *t* test was used to identify the difference in the normalized peak responses between the imagery condition and the baseline condition. The significance level was set at p<0.05.

# Additional information

## Funding

| Funder | Grant reference number | Author |
| --- | --- | --- |
| National Natural Science Foundation of China | 81790650 | Jia-Hong Gao |
| National Natural Science Foundation of China | 81790651 | Jia-Hong Gao |
| National Natural Science Foundation of China | 81727808 | Jia-Hong Gao |
| National Natural Science Foundation of China | 41830037 | Jia-Hong Gao |
| National Key Research and Development Program of China | 2017YFC0108901 | Jia-Hong Gao |
| Beijing Municipal Science and Technology Commission | Z171100000117012 | Jia-Hong Gao |
| Shenzhen Peacock Plan | KQTD2015033016104926 | Jia-Hong Gao |
| China Postdoctoral Science Foundation | 2018M641046 | Lingxi Lu |

| National Natural Science Foundation of China | 31421003 | Jia-Hong Gao |

The funders had no role in study design, data collection and interpretation, or the decision to submit the work for publication.

## Author contributions

Lingxi Lu, Conceptualization, Data curation, Formal analysis, Validation, Investigation, Visualization, Methodology, Writing—original draft, Project administration, Writing—review and editing; Qian Wang, Resources, Data curation, Formal analysis, Supervision, Methodology, Writing—review and editing; Jingwei Sheng, Software, Formal analysis, Validation, Methodology, Writing—review and editing; Zhaowei Liu, Data curation, Software, Writing—review and editing; Lang Qin, Visualization, Writing—review and editing; Liang Li, Conceptualization, Methodology; Jia-Hong Gao, Conceptualization, Resources, Data curation, Formal analysis, Supervision, Funding acquisition, Validation, Visualization, Methodology, Writing—original draft, Project administration, Writing—review and editing

## Author ORCIDs

Jia-Hong Gao https://orcid.org/0000-0002-9311-0297

## Ethics

Human subjects: For subjects in MEG experiment, written informed consents and consents to publish were obtained from all participants. The MEG experimental protocol was approved by the Peking University Institutional Review Board. For subjects in sEEG experiment, the written consent for participation was obtained, including permission for scientific publication. As one subject was a minor, written informed consent from his/her parent was obtained. The sEEG study was approved by the Ethics Committee of the Sanbo Brain Hospital, Capital Medical University.

## Decision letter and Author response

Decision letter https://doi.org/10.7554/eLife.48971.018
Author response https://doi.org/10.7554/eLife.48971.019

# Additional files

## Supplementary files

• Transparent reporting form DOI: https://doi.org/10.7554/eLife.48971.014

## Data availability

Source data files have been provided for Figures 1, 2, 4 and 5. MEG and sEEG data have been provided on the Open Science Framework under the identifier bacge (https://osf.io/bacge/).

The following dataset was generated:

| Author(s) | Year | Dataset title | Dataset URL | Database and Identifier |
|---|---|---|---|---|
| Lingx Lu, Qian Wang, Jingwei Sheng, Zhaowei Liu, Lang Qin, Liang Li, Jia-Hong Gao | 2019 | Speech mental imagery | https://osf.io/bacge/ | Open Science Framework, bacge |

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
