## [Decision Letter]

Thank you for submitting your article "Neural tracking of speech mental imagery" for consideration by *eLife*. Your article has been reviewed by three peer reviewers, and the evaluation has been overseen by a Reviewing Editor and Barbara Shinn-Cunningham as the Senior Editor. The following individual involved in review of your submission has agreed to reveal their identity: Thomas Cope (Reviewer #2).

The reviewers have discussed the reviews with one another and the Reviewing Editor has drafted this decision to help you prepare a revised submission.

The reviewers found the work interesting in demonstrating 'tagged' neural correlates using MEG and invasive recording of cyclical counting in premotor cortex, temporoparietal junction and occipitotemporal transition that could not be attributable to the stimulus.

Essential revisions:

1) The single most important comment is the attribution of the basis for the cyclical brain activity corresponding to the counting. This has a number of cognitive components including, for example, attention, working memory, phonological analysis, and (possibly) articulatory planning. To what extent do the authors feel that their use of the term 'speech mental imagery' is justified? The reviewers considered whether a more focused term was more appropriate. *eLife* does not encourage extensive further experiments or multiple cycles of revision that might address these components. What is required in a revision is a better justification of the term 'speech mental imagery' in the title and text or a reconsideration of this term.

2) The ANOVA with MEG is biased because the MEG activity focus was first defined by the contrast between the respective condition and baseline.

3) The independent repeated measures ANOVAs for each region is not the appropriate test here. Rather, an overall RM-ANOVA should be employed, with the test of interest being for a condition by region interaction. Post hoc tests should then be performed on marginal means, normalised for the overall effect of condition.

4) The activation of sensory-motor cortex is intriguing, but the claim that this area is involved in imagery speech as also claimed in (Bouchard et al., 2013) is not very compelling. For such a claim, it is necessary to ensure there was no articulatory movement (e.g. with EMG sensors). Otherwise, it could simply reflect a hardly noticeable yet existing muscle movement in the articulators as the subjects count the numbers.

---

## [Author Response]

Essential revisions:1) The single most important comment is the attribution of the basis for the cyclical brain activity corresponding to the counting. This has a number of cognitive components including, for example, attention, working memory, phonological analysis, and (possibly) articulatory planning. To what extent do the authors feel that their use of the term 'speech mental imagery' is justified? The reviewers considered whether a more focused term was more appropriate. eLife does not encourage extensive further experiments or multiple cycles of revision that might address these components. What is required in a revision is a better justification of the term 'speech mental imagery' in the title and text or a reconsideration of this term.

We completely agree with the view that several cognitive components were involved in the rhythmic inner counting task when participants were asked to construct their speech mental imagery. Mental imagery has been proposed to be essentially an attention-guided memory retrieval process, for example, in the visual domain (Ishai et al., 2002). Mental imagery of speech not only recruits the pathway by which phonological, lexical, or semantic details are retrieved from the memory system but also requires planned articulation that is simulated internally to enable individuals to have subjective experiences of their mental imagery (Tian and Poeppel, 2012; Tian et al., 2016). Therefore, these cognitive components cannot be definitely isolated from the term speech mental imagery.

In our study, neural tracking of speech mental imagery refers to the detection of the cyclical neural signal induced by the task of rhythmic inner counting. This cyclical neural signal can be attributed to the top-down regulation of the generation of number imagery, specifically, the internal arrangement of numbers into groups. Such internal organization is primarily based on the details of the inner representation of number imagery. In this way, the imagery-rate neural activity is tagged in the frequency domain and can be captured using neurophysiological measurements. It is known that the activation patterns of the brain network underlying speech mental imagery can be modulated by the task requirements and contextual variation (Tian and Poeppel, 2012, 2013; Tian et al., 2018). Therefore, the most important point to note is the operational definition of the internal organization of mental imagery, which is the key factor that induces cyclical neural signals. Considering that the term “speech mental imagery” as used in the previous manuscript is not focused enough to provide sufficient information about the intention and methodology of our study, we have now revised the term by adding the operational definition of the internal organization of mental imagery, namely, the task of rhythmic inner counting. Correspondingly, in the revised manuscript, we have added relevant content to justify the use of “speech mental imagery” as follows:

[Title] “Neural tracking of speech mental imagery during rhythmic inner counting”

[Discussion] “Mental imagery has been proposed to be essentially an attention-guided memory retrieval process, for example, in the visual domain (Ishai et al., 2002). […] The operational definition of speech mental imagery condition, namely, the rhythmic inner counting task, is not only related to the internal representations of speech but also requires high levels of working memory.”

Correspondingly, we have added the following references to the revised manuscript:

Ishai, Haxby and Ungerleider, 2002; Tian and Poeppel, 2012.

2) The ANOVA with MEG is biased because the MEG activity focus was first defined by the contrast between the respective condition and baseline.

ROI analysis between conditions can be biased when the ROIs are selected by contrasting between those conditions, leading to circular analysis that can distort the results (Kriegeskorte et al., 2009, Nature Neuroscience). To control the influence of circular analysis in our study, different statistical comparisons based on different null hypotheses were applied in ROI selection and ROI analysis. Specifically, we selected the ROIs from the significant activation maps, which were obtained by contrasting the whole brain activity at the target frequency bin and the average of its two neighbouring frequency bins *under each condition.* These activation maps illustrate the neural sources behind the rhythmic signals induced by mental imagery and auditory perception, respectively. The null hypothesis of the ROI selection was that there were no significant differences between the source-level activation at the target frequency bins and their neighbouring controllers. Different from the ROI selection criteria, we applied ANOVA on the ROI activation among three levels (at 0.8 Hz under the mental imagery condition, at 4 Hz under the mental imagery condition and at 4 Hz under the baseline condition) and 16 brain regions to identify the significant differences *between conditions*, which was done to quantitatively examine the activation difference in these brain regions between top-down and bottom-up processes. The null hypothesis for the ANOVA was that there were no significant differences in the neural activation among different conditions and brain regions, which differed from the assumption of the previous ROI selection criteria. In this way, we avoided circular analysis and reduced the distortion of our results caused by the ROI selection bias.

An important point to note is that, to some extent, the overlapping data subset applied for ROI selection and analysis might still cause bias due to the noise effect within the same data subset (Kriegeskorte et al., 2009). From this view, we agree with the comment that selection bias might enter into the ANOVA, despite the different statistical methods we have applied for ROI selection and analysis. We note that this statistical bias is not restricted to the current study; it potentially applies to any ROI analysis that attempts to compare ROI activity after defining the ROIs in the same data subset using non-independent ROI selection instead of independent selection criteria such as the anatomy atlas. Selection bias may cause overestimation of the results when comparing ROI data between conditions (Kriegeskorte et al., 2009). Critically, it should be noted that in our study, the results of ROI comparisons between conditions were *not* overestimated, as no condition difference was observed in the regions of IFG and SMG, which were the main neural clusters responsive for the top-down imagery construction according to source estimation, and the crucial findings were not based on the results of ROI analysis. However, we urge caution in interpreting the ROI results in the revised manuscript by adding the relevant content:

[Materials and methods] “Here, non-independent ROI selection criteria were chosen to quantitatively compare the source-level neural activation between conditions; thus, we interpreted the results with caution to avoid overestimation of the ROI data.”

3) The independent repeated measures ANOVAs for each region is not the appropriate test here. Rather, an overall RM-ANOVA should be employed, with the test of interest being for a condition by region interaction. Post hoc tests should then be performed on marginal means, normalised for the overall effect of condition.

As per the suggestion, we have conducted an overall repeated measures ANOVA (condition by brain region) for the ROI data. As a result, we found significant main effects of both condition and brain region and a significant interaction between the two variables. Then, we performed post hoc comparisons with Bonferroni correction on estimated marginal means to illustrate the differences between conditions for each brain region. We have revised the relevant content in the Results section as listed below:

“Two-way repeated measures ANOVAs (condition × brain region) for ROI activation showed a significant main effect of condition (F_2, 38_ = 23.56, *p* < 0.001, *ηp*^2^ = 0.55), a significant main effect of brain region (F_5.23, 99.41_ = 6.66, *p* < 0.001, *ηp*^2^ = 0.26), and a significant interaction between the two variables (F_8.09, 153.67_ = 5.60, *p* < 0.001, *ηp*^2^ = 0.23). […] No significant difference was observed in the brain activation levels at 4 Hz between the imagery condition and baseline condition, except in the right MFG, in which the activation at 4 Hz during mental imagery was stronger than that at baseline (*p* = 0.014).”

4) The activation of sensory-motor cortex is intriguing, but the claim that this area is involved in imagery speech as also claimed in (Bouchard et al., 2013) is not very compelling. For such a claim, it is necessary to ensure there was no articulatory movement (e.g. with EMG sensors). Otherwise, it could simply reflect a hardly noticeable yet existing muscle movement in the articulators as the subjects count the numbers.

We fully agree with the comment on the point that it is necessary to exclude the influence of muscle movements to reach a compelling conclusion on the role of the sensorimotor cortex in speech mental imagery. We respond to two aspects of this helpful comment:

First, Bouchard et al. (2013) claimed that speech-articulator representations are arranged somatotopically in the ventral PrG and PoG during syllable production, while our study found that the ventral PrG and PoG were engaged in constructing speech imagery when participants were told to count numbers in their mind without moving their lips, jaw or tongue. Our result is consistent with previous findings concerning the involvement of the sensorimotor cortex in speech imagery. For example, Tian et al. (2016) reported that the sensorimotor cortex showed stronger activation in articulatory imagery in which participants generated kinaesthetic feelings of articulation than in hearing imagery in which motor-related imagery was discouraged, suggesting that speech mental imagery with motor-related feelings could activate the sensorimotor cortex. In addition, the neural signals directly recorded from the sensorimotor cortex contributed to the imagined-speech decoding in both hearing imagery without kinaesthetic representation (Martin et al., 2016) and silent reading of speech (Martin et al., 2014). We have now revised the statement and have drawn a more careful conclusion about the involvement of the sensorimotor cortex.

Second, an earlier study by Kleber et al. (2007) attempted to record the EMG sensor placed on the throat and ensured that the muscle movements in imagined singing were significantly weaker than those in overt singing and did not differ from those during the resting state. After taking EMG data into consideration, Kleber et al. (2007) still found activation in the sensorimotor cortex in imagined singing, which could not be accounted for by muscle movements in articulators. However, to our knowledge, most imagery studies have not strictly controlled for articulator movement using EMG recording (Tian et al., 2016; Martin et al., 2014, 2016) but primarily based the results on the assumption that participants followed the instructions on how to generate mental imagery. This valuable comment raises the importance of controlling the articulatory movement effect, which should be noted in future studies, especially when the activation of the sensorimotor cortex is observed in mental imagery studies.

Thus, to clarify the logic and to address the concerns about the effect of muscle movements, we have revised the relevant content in the Discussion section as follows:

“In addition, our study shows that the ventral PrG and PoG are also activated during the construction of speech imagery, which is consistent with previous findings that stronger activation of the sensorimotor cortex induced by articulatory imagery than hearing imagery (Tian et al., 2016) and the direct neural signal from the sensorimotor cortex contributed to the decoding of imagined speech and silent reading (Martin et al., 2014, 2016). Nevertheless, a firm conclusion on the contribution of the sensorimotor cortex to the construction of speech mental imagery will only be possible after controlling for any articulator movement in future studies.”

Correspondingly, we have added relevant references to the manuscript:

Martin et al., 2016; Martin et al., 2014.